# Quality of family relationships and outcomes of dementia: a systematic review

Hannah B Edwards,[1,2] Sharea Ijaz,[1,2] Penny F Whiting,[1,2] Verity Leach,[1,2] Alison Richards,[1,2] Sarah J Cullum,[3] Richard IL Cheston,[4] Jelena Savović[1,2]

HBE and SI are joint first authors.

[1]Bristol Medical School, University of Bristol, Bristol, UK
[2]National Institute for Health Research (NIHR) Collaboration for Leadership in Applied Health Research and Care (CLAHRC) West, University Hospitals Bristol NHS Foundation Trust, Bristol, UK
[3]Psychological Medicine, School of Medicine, Faculty of Medical and Health Sciences, University of Auckland, Auckland, New Zealand
[4]Department of Health and Social Sciences, University of the West of England, Bristol, UK

**Correspondence to**
Dr Jelena Savović;
j.savovic@bristol.ac.uk

## ABSTRACT

**Objectives** To evaluate the association between the quality of relationship between a person with dementia and their family carer and outcomes for the person with dementia.

**Design** Systematic review.

**Eligibility criteria** Cohort studies of people with clinically diagnosed dementia and their main carers. Exposures of interest were any elements of relationship quality, for example, attachment style, expressed emotion and coping style. Our primary outcome was institutionalisation, and secondary outcomes were hospitalisation, death, quality of life and behavioural and psychiatric symptoms of dementia ('challenging behaviour').

**Data sources** MEDLINE, Embase, Web of Science, PsycInfo, the Cochrane Library and Opengrey were searched from inception to May 2017.

**Study appraisal and synthesis methods** The Newcastle-Ottawa Scale was used to assess risk of bias. A narrative synthesis of results was performed due to differences between studies.

**Results** Twenty studies were included. None of the studies controlled for all prespecified confounding factors (age, gender, socioeconomic status and severity of dementia). Reporting of results was inadequate with many studies simply reporting whether associations were 'statistically significant' without providing effect size estimates or CIs. There was a suggestion of an association between relationship factors and global challenging behaviour. All studies evaluating global challenging behaviour provided statistical evidence of an association (most P values below 0.02). There was no consistent evidence for an association for any other outcome assessed.

**Conclusions** There is currently no strong or consistent evidence on the effects of relationship factors on institutionalisation, hospitalisation, death or quality of life for people with dementia. There was a suggestion of an association between relationship factors and challenging behaviour, although the evidence for this was weak. To improve our ability to support those with dementia and their families, further robust studies are needed.

**PROSPERO registration number** CRD42015020518.

## Strengths and limitations of this study

► Broad search strategy so unlikely to have missed relevant studies.
► Double screening minimises selection bias.
► We were not able to assess publication bias and the potential for selective reporting of outcomes within studies.

## BACKGROUND

Dementia is a key public health concern in the UK[1–3] with around 7% of all those over 65 affected, and the numbers of people with dementia predicted to double every 20 years.[4–6] Institutionalisation (being placed in a full-time care/nursing home) is a key outcome for people with dementia, their families and the healthcare system. Although in many circumstances institutionalisation may be the best or only option for the person affected by dementia, most people report that they would prefer to stay living in their own home.[7] Recent media attention to a few very poorly run care homes has also led to concerns about institutionalisation.[8–10] Additionally, the financial cost of full-time care is very high, both for affected individuals and their families, and for the public, as public taxes are used to contribute to care home fees. Consequently, for some time it has been UK government policy to help families to continue supporting people with dementia at home, specifically to delay or avoid institutionalisation.[11]

The quality of relationship between the person with dementia and people who care for them has been linked with a range of outcomes including institutionalisation, cognitive and functional decline and quality of life (QoL).[12–21] There is also growing interest in the potential for psychosocial interventions to improve outcomes by enhancing interactions within families.[22 23] If families are better equipped to cope with the psychological and emotional challenges of dementia, then care at home may be sustained for longer. However, it is not clear which elements of the

relationship are predictive of early institutionalisation or which lead to a faster decline. This evidence is necessary both to justify and to help to develop early psychosocial interventions, ideally at the point of diagnosis.

To address this issue, we performed a systematic review of the evidence on how elements of relationship quality between the person with dementia and their main informal (family) carer are associated with outcomes for the person with dementia.

## METHODS
This study was a systematic review, registered with the PROSPERO International Prospective Register of Systematic Reviews, registration number CRD42015020518. The full protocol has been published.[24]

### Eligibility criteria
Only cohort studies (prospective and retrospective) were included in the review. Relevant systematic reviews were obtained and used as a means of identifying other original studies.[24] Qualitative, case–control (unless nested in a prospective cohort), and cross-sectional studies were excluded.

The population of interest was people with dementia and their main informal caregiver (most commonly a spouse or child). Professional paid caregivers were excluded. People with all types of clinically diagnosed dementia were included.

The exposures of interest were factors that capture an element of relationship quality. We adopted a broad definition of 'relationship quality' as how happy or satisfied an individual is in their relationship.[25] Attachment style, coping style, affection and expressed emotion (EE) were all identified at the design stage as key exposures of interest. While affection is a relatively straightforward term, attachment style, coping and EE relate to specific psychological constructs. 'Attachment style' is a term originally developed to understand the emotional relationship between children and parents but has since been extended to adult romantic relationships. Four main styles of attachment have been identified in adults: secure, anxious–preoccupied, dismissive–avoidant and fearful–avoidant.[26 27] Coping is a wide-ranging construct that includes elements that are clearly measures of relationship quality (eg, 'relationship-focused coping') and those that are more individual in nature (eg, acceptance coping). However, as even individual coping styles are typically initiated in response to aspects of relationships, we felt that this was an appropriate exposure to capture. 'EE' is a measure of the family environment based on how the relatives of a psychiatric patient spontaneously talk about/to the patient. High levels of EE have been associated with worse prognosis in a number of mental illnesses including schizophrenia. This may be due to emotional overinvolvement, which can be experienced as hostility, criticism and intolerance.[28 29]

Two other notable factors emerged as key exposures when reviewing the literature: 'mutuality' and 'boundary ambiguity'. Mutuality is a cluster concept capturing levels of positive engaging interaction, attachment and emotional support. Boundary ambiguity involves uncertainty about whether a person is in or out of the family group. This occurs as a result of significant changes in that person including those cognitive, functional, mood and personality changes that are indicative of dementia. Boundary ambiguity is associated with and taken as an indicator of emotional distancing and the withdrawal of the caregiver from the person with dementia. Other factors emerging from the literature were included if they captured an element of relationship quality, and this was assessed on a case-by-case basis through discussion with the study team. Carer abuse was excluded, as this was considered to be a different area of research. Overall measures of carer burden were also excluded as an exposure because they relate more to cognitive and functional levels in dementia than to relationship quality.

The primary outcome of interest was institutionalisation. This is a key event in the course of dementia, both socially and financially, from the perspectives of the individual, their family and the public. Secondary outcomes were hospitalisations, death (or time to death), QoL and challenging behaviour (also referred to as the behavioural and psychological symptoms of dementia or BPSD). Examples of challenging behaviour can include depression, anxiety, aggression, paranoia, hallucinations and delusions. Studies measuring QoL or challenging behaviour as the outcome had to use validated assessment tools to be included in the review.

### Search strategy
MEDLINE, Embase, Web of Science, PsycInfo, the Cochrane Library and Opengrey were searched from inception to May 2017 without any language restrictions. The full search strategy is available in the supplementary data of our published protocol.[24] All results were imported into an Endnote X7 reference library and into a bespoke-built Microsoft Access 2013 database to manage screening.

### Selection of papers
The titles and abstracts of all identified papers were screened in duplicate by two reviewers working independently, and all potentially relevant papers were retrieved. All retrieved papers were read in full and assessed for eligibility using a standardised and piloted inclusion checklist, applied by two reviewers independently. Any discrepancies between the reviewers (at either stage of screening) were resolved through discussion.

### Data extraction
Data were extracted from included studies using a bespoke data collection form, which was piloted on six studies and amended as a result of the piloting. Data extracted included study characteristics, characteristics of

the population of people with dementia and their carers, recruitment and response, the exposures and outcomes and how and when they were measured, and details of the analyses and results. The terminology for risk factors and outcomes was used as reported by the original study authors. Where numerical results were incompletely reported, where possible, relevant results (effect estimates, SE and 95% CIs) were calculated from the raw data. For continuous exposures, effect sizes were presented as change in the outcome for a one unit increase in the exposure. Multiple publications of the same dataset were counted as a single study. In the case where analyses of the same association was repeated in more than one report, our policy was either to include the result based on the largest sample size only, or if sample sizes were equivalent, then we would include the result from the most recent publication only. Data were extracted from the published reports only; we did not contact authors for additional unpublished information.

## Analysis

We planned to use meta-analysis to estimate summary effect sizes if there had been sufficient studies with similar populations, exposures and outcomes. As meta-analysis was not possible, a narrative synthesis of results is provided.[24]

## Assessment of methodological quality and risk of bias

The Newcastle-Ottawa Scale[30] (NOS) was used to assess risk of bias for included reports. This is an eight-item questionnaire that assesses the following methodological criteria: representativeness of the exposed cohort; selection of the non-exposed cohort; ascertainment of exposure; demonstration that the outcome of interest was not present at the start of the study; comparability of cohorts (risk of confounding); assessment of exposure and outcome; and length and adequacy of follow-up. NOS allocates 'stars' for adequate methods but does not specifically advise calculating the sum of allocated stars to give an overall score.[30] Empirical evidence also suggests that numerical quality scores are not helpful in differentiating between studies of high and low risk of bias.[31] For this reason, we considered each of the eight criteria of the NOS tool separately and assessed the study as having adequate methods for that particular aspect of study conduct if the 'star' could be allocated for that NOS criterion. In our protocol, we considered 10 factors as potentially important confounding domains.[24] During the piloting of the data extraction and risk of bias assessment, the team agreed a minimum number of essential confounders that all studies should have adjusted for. A study had to control for the following four prespecified factors to be at low risk of confounding: age, gender, socioeconomic status (SES) and dementia severity. We also recorded other confounders studies had adjusted for, in addition to the four main confounders used for risk of bias assessment.

## RESULTS

The search identified 9321 potentially relevant papers. Of these, 190 papers were retrieved for full-text screening, 23 publications[20 32–53] met the eligibility

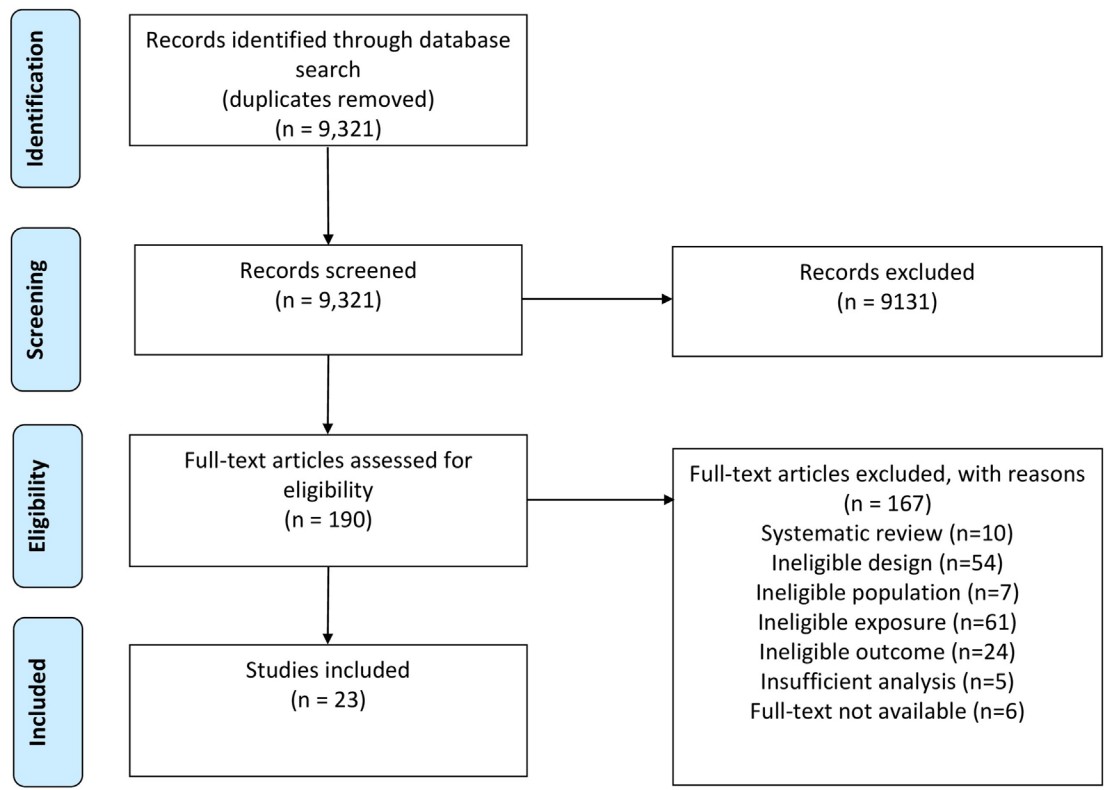

**Figure 1** PRISMA flow diagram. PRISMA, Preferred Reporting Items for Systematic Review and Meta-Analysis.

criteria (figure 1). Four of the 23 publications were based on data from one cohort study[32–35] and so the total number of unique studies was 20 (2340 participants with dementia). As the four connected publications each contributed unique results, all were included in the review.

## Study characteristics

The majority of included unique studies (14 out of 20) came from the USA, and there was one each from the UK, Canada, Switzerland, Belgium, the Netherlands and Australia. Sample sizes in the relevant analyses ranged from 29 to 220 (mean 143). Years of publication ranged from 1990 to 2016, with most from the 1990s and early 2000s. The most frequently reported dementia type was Alzheimer's disease, although in eight studies, the distribution of dementia types was not reported. Time since diagnosis ranged from 3 months to 6.5 years, although in six studies, this was not reported. The majority (13/20) did not report participants' ethnicity, where this was reported, cohorts were predominantly Caucasian. The majority of caregivers included were spouses (100% in seven studies and more than 50% in another seven studies). Other carers were children of the person with dementia. The characteristics of included studies are presented in table 1.

## Risk of bias in included studies

None of the studies met all eight NOS criteria. All studies had adequate ascertainment of exposure as these were all based on structured interviews. All but one also had adequate 'demonstration that the outcome of interest was not present at start of study' and 'follow up long enough for outcome to occur'. Blind independent outcome assessment was not possible in many studies with institutionalisation as the outcome, because this outcome tended to be self-reported by the carer table 2.

In total, 60 separate analyses were included in this review, of which 40 (two-thirds) did not control for any potential confounding factors. In the 20 analyses that did include some adjustment, only a minority adjusted for any of the factors we identified as key potential confounders. None of the studies adjusted for all four key confounding factors. Three studies each adjusted for three out of the four key confounding factors.[33 50 53] A further two studies adjusted for two of the four key confounders.[46 47]

Seven studies had inadequate follow-up of the cohort (loss to follow-up).[37 47 48 50–53] Many studies presented insufficient information to make a clear judgement on some of the NOS criteria. In addition to potential risk of bias, many studies had reporting problems. Of the 60 included results, eight (13%) neither reported effect size estimates nor CIs, 40 (67%) reported an effect size with no CIs and 23 (38%) did not report specific P values. Only six analyses (10%) fully reported their results with effect size, CIs and P values.

## Primary outcome: risk factors for institutionalisation

Ten studies examined 25 different relationship quality factors as potential risk factors for institutionalisation. Follow-up ranged from 6 to 24 months. The majority of studies found no association between the risk factors investigated and the incidence of institutionalisation. Although some individual studies reported associations between relationship quality and institutionalisation, there were no consistent findings across risk factors, and the lack of appropriate adjustment for basic confounding factors makes interpreting the results very difficult table 3.

## Risk factors for challenging behaviour (BPSD)

Under the umbrella term of challenging behaviour four main types of outcome were evaluated: global challenging behaviour scores, psychotic symptoms, depression and other BPSD outcomes. Eight studies examined nine different relationship quality factors as potential risk factors for these aspects of challenging behaviour. The length of follow-up ranged from 6 to 24 months table 4.

There was a suggestion of an association between relationship factors and global challenging behaviour. All studies evaluating global challenging behaviour provided statistical evidence of an association (most P values below 0.02). However, one study that reported two analyses did not report effect sizes.[46] For another, the reported effect size was very small (mean difference of 0.23 on a scale of 1–144).[41] A larger effect size was seen for the association between EE and global challenging behaviours (mean difference of 1.9 in a scale of 0–8).[44] None of these analyses adjusted for our prespecified confounding factors.

Most studies found no evidence of an association between relationship factors and either psychotic symptoms, depression or other BPSD outcomes. However, some of these were small studies that may have been underpowered to detect an association. One study adjusted for three out of four prespecified potential confounders.[33] This was also one of the largest studies (n=171). It found no evidence for an association between couple mutuality and psychotic symptoms and a very weak, likely clinically negligible effect of this factor on depression (mean difference −0.43 points on a scale of 0–68).[33]

## Risk factors for hospitalisation, QoL and death

The outcomes of hospitalisation[35] were examined in one study, and QoL[37 52] and death[51 53] were each examined in two studies. The small number of poor quality studies means it is not possible to draw conclusions regarding the association of relationship factors with these outcomes table 5.

## DISCUSSION

This systematic review assessed the evidence on the role of relationship factors on outcomes in dementia. Although it is plausible that relationship factors could affect the risk of institutionalisation, challenging behaviour and other outcomes, there is currently no robust evidence to

**Table 1** Characteristics of included studies

| Study ID | Country | People with dementia | | | | | | Carers | | | | | Exposures* | Outcomes* |
|---|---|---|---|---|---|---|---|---|---|---|---|---|---|---|
| | | n | % Female | Dementia type | Dementia duration years (SD); range | Dementia severity Mean (SD); range | Age years (SD); range | n | % Female | % Spouse | % Child | Age years (SD); range | | |
| Caron[36] | USA | 60 | 22 | AD | 3.4 (2.5); 1–13 | NR | 68.26 (8.44); 53–90 | 60 | 79 | 86 | 8 | 63.57 (8.94); 30–84 | Boundary ambiguity (caregiver closeout) | BPSD – depression; anxiety; paranoia; activity disturbance |
| Clare et al[37] | UK | 51 | 50.98 | 61% AD, 21% VD, 17% mixed | NR | MMSE=24.5 (2.80); 18–30 | 76.75 (7.88); 55–91 | 51 | 70 | 57 | 31 | 65.06 (14.50); 33–89 | Positive quality of relationship: Positive Affect Index | QoL Alzheimer's Disease Scale |
| Kunik et al (four papers)[32–35] | USA | 215 | 4.6 | NR | ≤1 | NR | 76 (6.2) | 215 | NR | NR | NR | NR | Mutuality, relationship strain | BPSD – aggression, depression, psychosis, hospitalisation |
| Spruytte et al[38] | Belgium | 144 | 68.8 | NR | ≥0.25 | GDS mean=6 | 82; 61–94 | 144 | 69.4 | 38.9 | 50.7 | 63; 38–90 | Quality of relationship; criticism | Institutionalisation |
| Fisher and Lieberman[39] | USA | 164 | 0.44 | NR | 6.5 | MMSE=17 (6.25) | 77 (7) | 164 | 54% non-NHP, 47% NHP | 0 | 100 | 45 (9) | Emotional closeness; boundary ambiguity; negative family feelings | Institutionalisation |
| Burgener and Twigg[20] | USA | 96 | 57 | NR | ≤1 | MMSE=21.52; 10–29 | 77.3 (7.8); 55–96 | 96 | 74 | 56 | 40 | 63.7 (12.2); 35–83 | Quality of relationship | Institutionalisation |
| Stevens et al[40] | USA | 215 | 71.2 | NR | NR | MMSE=12.2 (8.7) | 74.4 (8.2) | 215 | NR | 47.9 | 40.5 | NR | Coping strategies | Institutionalisation |
| Perren[41] | Switzerland | 68 | NR | 60% AD, 25% VD, 15% other | NR | MMSE=21.6 (5.1) | 74.8 (7.6) | 68 | NR | 100 | 0 | 70.9 (10.0) | Attachment style | Overall care recipients problem behaviour |
| Wright[42] | USA | 29 | 50 | AD | 4.8; 1–11 | MMSE=18; 4–22, GDS=4.4; 2–6 | 67.5; 51–83 | 30 | 80 | 100 | 0 | 67.4; 51–81 | Cohesion; tension; affection; present/past marital happiness | Institutionalisation |
| Wright et al[43] | USA | 14 | 50 | AD | 2.8 | MMSE of 21.29 (3.69); 16–26 | 65; 49–85 | 14 | 50 | 100 | 0 | NR | Cohesion; tension; affection | BPSD – depression |
| Vitaliano et al[44] | USA | 77 | 32 | AD | 4.3 (2.1) | NR | 70.9 (6.9) | 77 | NR | 100 | 0 | 67.2 (7.4) | Expressed emotion | Depression; negative behaviour |
| Bannister et al[45] | UK | 116 | 72.4 | 72% probable AD, 17% VD, 10% LB, 3% unclear | NR | CAMCOG mean=45.2 | 79.8 | 116 | 59.5 | 14.7 | NR | 65 | Coping strategies | Institutionalisation |
| de Vugt et al[46] | The Netherlands | 99 | 56.6 | 74% AD, 19% VD, 2% frontal lobe, 3% Parkinson's, 2% mixed | 3.5 (31.1) | MMSE=18.2 (4.8) | 78.2 (8.4) | 99 | 66.6 | 55.6 | 44.4 | 61.9 (11.9) | Coping strategies | Psychosis; mood/apathy; hyperactivity; overall BPSD |

Continued

**Table 1** Continued

| Study ID | Country | People with dementia | | | | | | Carers | | | | | Exposures* | Outcomes* |
|---|---|---|---|---|---|---|---|---|---|---|---|---|---|---|
| | | n | % Female | Dementia type | Dementia duration years (SD); range | Dementia severity Mean (SD); range | Age years (SD); range | n | % Female | % Spouse | % Child | Age years (SD); range | | |
| Wells and Over[47] | Australia | 93 | 42 | NR | 5.6 (4.2) | NR | 76.1 (7.3) men; 74.6 (6.2) women | 93 | 58 | 100 | 0 | 74.7 (6.2) men; 71.4 (7.9) women | Coping strategies | Institutionalisation |
| Torossian and Ruffins[48] | USA | 197 | 60.9 | AD | NR | GDS range=4-7 | NR | 197 | 39.1 | 100 | 0 | 70.7 | Adaptability; cohesion | Institutionalisation |
| Markiewicz et al[49] | Canada | 108 | 56.6 | 92% AD, 8% dementia+stroke | 3.43 (2.57) | MMSE=12.86 (7.95) | 74.42 (8.37) | 108 | 68.1 | 65.5 | 34.5 | 62.24 (12.97) | Attachment styles | Institutionalisation |
| Pruchno et al[50] | USA | 220 | NR | AD | NR | NR | NR | 220 | 67.9 | 100 | 0 | 70.2; 45–94 | Quality of relationship | Institutionalisation |
| McClendon et al[51] | USA | 141 | 45 | 71.5% probable AD, 28.5% possible AD | 4.14 (2.47) | MMSE=17.28 (6.50) | 72.46 (7.91) | 141 | NR | 75 | NR | NR | Caregiver coping | Time to death |
| Shroff[52] | USA | NR | NR | NR | NR | NR | NR | 83 | 85.5 | 54.3 | NR | 62.38 (18.21) | Family coping and coherence | PwD QoL |
| Snyder[53] | USA | 233 | 56.7 | NR | 3.49 (1.80) | NR | 86.08 (5.83) | 233 | 78.5 | 37.3 | 53.6 | 66.27 (13.21) | Carer coping strategies | Incidence of severe dementia, time to institutionalisation, mortality |

*The exposures and outcomes listed in the table refer only to the ones relevant to our review question.
Study design: all studies were prospective cohorts except for Godwin (randomised trial) and Scroff (controlled before and after intervention study), for these two studies we ignored the intervention status.
AD, Alzheimer's dementia; BPSD, Behavioural and Psychological Symptoms of Dementia; CAMCOG, Cambridge Cognition score; GDS, Global Deterioration Scale; LB, Lewy body dementia; MMSE, Mini-Mental State Examination; n, numbers in analysis; NHP, nursing home placement; NR, not reported; PwD, person with dementia; QoL, quality of life; VD, vascular dementia.

**Table 2** Risk of bias assessment based on Newcastle-Ottawa Scale

| Study Author Year | Country | Newcastle-Ottawa Scale items addressing different methodological components of the study and potential sources of bias | | | | | | | |
|---|---|---|---|---|---|---|---|---|---|
| | | Representativeness of the exposed cohort | Selection of the non-exposed cohort | Ascertainment of exposure | Outcome of interest was not present at start of study | Comparability of cohorts on the basis of the design/analysis (confounding) | Assessment of outcome blind/ record linkage | Follow-up long enough for outcome to occur | Adequacy of follow-up of cohorts (attrition) |
| Pruchno et al 1990[50] | USA | ✓ | ✓ | ✓ | ✓ | ✗* | ✗ | ✓ | ✗ |
| Vitaliano et al 1993[44] | USA | ✓ | ✓ | ✓ | ✓ | ✗ | ✗ | ✓ | ✓ |
| Wright 1994[42] | USA | ✓ | ✓ | ✓ | ✓ | ✗ | ✓ | ✓ | ✓ |
| Markiewicz et al 1997[49] | Canada | ✓ | ✓ | ✓ | ✓ | ✗ | ✓ | ✓ | ✓ |
| Bannister et al 1998[45] | UK | ✓ | ✓ | ✓ | ✓ | ✗ | ✗ | ✓ | ✓ |
| Wells and Over 1998[47] | Australia | ✓ | ✓ | ✓ | ✓ | ✗ | ✗ | ✓ | ✗ |
| Wright et al 1998[43] | USA | ✗ | ✗ | ✓ | ✓ | ✗ | ✗ | ✓ | ✓ |
| Caron et al 1999[36] | USA | ✗ | ✗ | ✓ | ✓ | ✗ | ✗ | ✓ | ✓ |
| Fisher and Lieberman 1999[39] | USA | ✓ | ✓ | ✓ | ✓ | ✗ | ✗ | ✓ | ✓ |
| Torossian and Ruffins 1999[48] | USA | ✗ | ✓ | ✓ | ✓ | ✗ | ✗ | ✓ | ✗ |
| Spruytte et al 2001[38] | Belgium | ✓ | ✓ | ✓ | ✓ | ✗ | ✗ | ✗ | ✓ |
| Burgener and Twigg 2002[20] | USA | ✓ | ✓ | ✓ | ✓ | ✗ | ✗ | ✓ | ✓ |
| de Vugt et al 2004[46] | Netherlands | ✓ | ✓ | ✓ | ✗ | ✗ | ✗ | ✓ | ✓ |
| McClendon et al 2004[51] | USA | ✗ | ✓ | ✓ | ✓ | ✗ | ✓ | ✓ | ✗ |
| Stevens et al 2004[40] | USA | ✓ | ✓ | ✓ | ✓ | ✗ | ✗ | ✓ | ✓ |
| Perren et al 2007[41] | Switzerland | ✓ | ✓ | ✓ | ✓ | ✗ | ✗ | ✓ | ✓ |
| Kunik et al 2010 (four papers)[32–35] | USA | ✓ | ✓ | ✓ | ✓ | ✗* | ✗ | ✓ | ✓ |
| Clare et al 2014[37] | UK | ✗ | ✓ | ✓ | ✓ | ✗ | ✗ | ✓ | ✗ |
| Shroff 2015[52] | USA | ✓ | ✓ | ✓ | ✓ | ✗ | ✗ | ✗ | ✗ |
| Snyder 2016[53] | USA | ✓ | ✓ | ✓ | ✓ | ✗* | ✓ | ✓ | ✗ |

✓=study dealt with this adequately. ✗=study was at risk of bias in this area, or provided no information to demonstrate otherwise.
*These studies adjusted for three out of the four key confounders.

**Table 3** Associations between relationship factors and institutionalisation (10 studies)

| Risk factor | Study | n | Follow-up | Results (95% CI; P value) | Analysis adjusted for* |
|---|---|---|---|---|---|
| Factors relating to the interaction between the person with dementia and their caregiver | | | | | |
| Quality of relationship | Spruytte et al[38] | 144 | 6–9 m | OR 0.92 (P=0.02) | NR |
| | Pruchno et al[50] | 220 | 12 m | OR 1.31 (P>0.05) | a, g, s, r, t, c, bp, ADL, ci, m, hs, cb |
| Marital happiness At baseline | Wright[42] | 29 | 24 m | PoV 0.313 (P<0.01) | NR |
| Before dementia onset | | | | PoV 0.045 (P>0.05) | NR |
| Emotional closeness | Fisher and Lieberman[39] | 164 | 24 m | OR 1.64 (1.09, 2.46; P=0.02) | d, ba, nf, oc, fe |
| Emotional distancing/boundary ambiguity | Fisher and Lieberman[39] | 164 | 24 m | OR 1.25 (0.85, 1.83; P=0.26) | d, ba, nf, oc, fe |
| | Wells and Over[47] | 93 | 12–18 m | OR 1.3 (P>0.05) | a, d, t |
| Cohesion | Torossian and Ruffins[48] | 197 | 24 m | MD NR (P>0.05) | NR |
| | Wright[42] | 29 | 24 m | PoV 0.472 (P<0.001) | NR |
| Affection | Wright | 29 | 24 m | PoV 0.144 (P>0.05) | NR |
| Warmth† | Spruytte et al[38] | 144 | 6–9 m | NR (P<0.05) | NR |
| Anxious-ambivalent attachment (carer) | Markiewicz et al[49] | 108 | 24 m | OR 1.27 (P=0.36) | NR |
| Avoidant attachment (carer) | Markiewicz et al[49] | 108 | 24 m | OR 2.39 (P<0.001) | NR |
| | Stevens et al[40] | 215 | 24 m | HR 1.011 (P=0.37) | NR |
| High levels of tension | Wright[42] | 29 | 24 m | PoV 0.162 (P>0.05) | NR |
| Excessive criticism*† | Spruytte et al[38] | 144 | 6–9 m | NR (P>0.05) | NR |
| Factors mainly relating to the caregiver | | | | | |
| Limited adaptability | Torossian and Ruffins[48] | 197 | 24 m | MD NR (P>0.05) | NR |
| Approach coping | Stevens et al[40] | 215 | 24 m | HR 0.997 (P=0.77) | NR |
| Directing relative's behaviour | Bannister et al[45] | 116 | 12 m | MD 0.2 (P=0.40) | NR |
| Keeping relative busy | Bannister et al[45] | 116 | 12 m | MD 0.5 (P=0.02) | NR |
| Learning about the illness | Bannister et al[45] | 116 | 12 m | MD 0.3 (P=0.42) | NR |
| Prioritising | Bannister et al[45] | 116 | 12 m | MD 0.1 (P=0.52) | NR |
| Reducing expectations | Bannister et al[45] | 116 | 12 m | MD 0.4 (P=0.19) | NR |
| Consistent larger sense of the illness | Bannister et al[45] | 116 | 12 m | MD 0.4 (P=0.35) | NR |
| Positivity | Bannister et al[45] | 116 | 12 m | MD 0.5 (P=0.57) | NR |
| | Wells and Over[47] | 93 | 12–18 m | OR 1.03 (P>0.05) | a,d |
| Seeking social support | Wells and Over[47] | 93 | 12–18 m | OR 1.91 (P<0.05) | a,d |
| | Snyder[53] | 233 | NR | HR 1.159(0.718 to 1.87; P=0.55) | a, d, g, nc |
| Accepting responsibility | Wells and Over[47] | 93 | 12–18 m | OR 0.28 (P<0.01) | a, d |

Continued

**Table 3** Continued

| Risk factor | Study | n | Follow-up | Results (95% CI; P value) | Analysis adjusted for* |
|---|---|---|---|---|---|
| Confrontational | Wells and Over[47] | 93 | 12–18m | OR 2.12 (P<0.05) | a, d |
| Negative feelings | Fisher and Lieberman[39] | 164 | 24m | OR 1.47 (0.99 to 2.19; P=0.05) | d, ba, oc, ec, fe |

*Prespecified key confounders: a, age; d, dementia severity; g, gender; s, socioeconomic status. All other confounders: ADL, activities of daily living; ba, boundary ambiguity; bp, behaviour problems; c, number of children; cb, caregiver burden; ci, caregiver illness; ec, emotional closeness; fe, family efficiency; hs, help services used; m, medication; nc, non-coresidency; nf, negative feeling; oc, organised cohesiveness; r, religion; t, time spent (duration of caregiving).

†These are the subscales of the relationship quality scale.

HR >1 indicates increased risk, HR <1 indicates decreased risk. OR >1 indicates increased odds of outcome, OR <1 indicates decreased odds of outcome.

Corr, correlation (This is a number between −1 to +1 and indicates the degree to which the exposure and outcome vary together. Positive numbers indicate that exposure and outcome increase together, while negative numbers indicate that the outcome increases as the exposure decreases. Larger numbers indicate stronger correlation); MD, mean difference in outcome (Large differences between groups suggests that the exposure might affect the outcome. For continuous exposures, mean difference represents the change in the outcome for one unit increase in the exposure.); NR, information not reported (where possible results and 95% CIs were calculated from raw data); PoV, proportion of variance (This is a number between 0–1 indicating the proportion of variance in the outcome explained by the exposure. Higher numbers indicate greater explanatory power of the exposure.)

establish which, or to what extent, elements of the caring relationship affect specific outcomes.

The majority of studies found no association between specific risk factors and the outcomes of interest. However, this is not necessarily evidence of a lack of association, as some of the studies will have been underpowered to detect differences, and most were at risk of confounding as they failed to adjust for even the most basic confounding factors. There was a suggestion of an association between factors related to the emotional withdrawal of the caregiver and subsequent increased risk of challenging behaviour in the person with dementia. All studies in this category found statistical evidence of an association. However, the methodological quality of these studies was poor. For example, many did not report effect sizes, while in others the effect sizes were small, suggesting that associations may not be clinically important. This could also be because the sample was too small to detect a difference. No study reported justification for their sampling or the sample sizes. There was also a potential for confounding in studies that did report effect sizes.

One of the strengths of this review is a thorough, sensitive search that will have minimised the chance of missing relevant studies. By limiting inclusion to cohort studies only, we avoided interpretive difficulties from recall bias (systematic differences in how people remember risk factors) and reverse causation (when a purported risk factor is in fact a result of the outcome). The double-screening of each record by two reviewers working independently also minimised the possibility of errors and selection bias in the identification of eligible reports. All data extracted was checked in full by a second reviewer.

A limitation of our review is that we were not able to assess for publication bias or selective reporting of results. It is known that studies with null findings are less likely to be published, and authors can selectively report their 'more interesting' findings.[54] This means it is possible there are other relevant results that have not been reported and so would not be available to a review. We attempted to minimise the risk of publication bias by including a grey literature search.

Our ability to answer the research question was limited due to unclear and incomplete reporting, few studies assessing similar risk factors and the potential for bias in included studies. We considered the appropriateness of using meta-analysis to produce summary estimates. However, many studies did not report full details of effect sizes and CIs that are required for meta-analysis. Additionally, most risk factor–outcome relationships were only reported in one or two studies with conflicting results or non-comparable outcome measures between studies), thus the meta-analyses would have been inappropriate. It would not have been meaningful or appropriate to combine different risk factors or outcomes in pooled analysis. We assessed the potential for bias in included studies and took these assessments into consideration

**Table 4** Associations between relationship factors and challenging behaviour (BPSD) (eight studies)

| Risk factor | Study | n | Follow-up | Results (95% CI; P value) | Analyses adjusted for* | Outcome assessment tool, scale range |
|---|---|---|---|---|---|---|
| **Outcome: global challenging behaviour (BPSD) or problem behaviour (four studies)** | | | | | | |
| Emotional distancing/boundary ambiguity | Caron et al[36] | 60 | 12 m | Corr 0.27 (P<0.001) (overall) | NR | BEHAVE-AD, 0–75 |
| Non-adaptive coping | de Vugt et al[46] | 69 | 6 m | MD NR (P=0.007) | g, s, ct | NPI, 1–144 |
| | | 69 | 12 m | MD NR (P=0.019) | g, s, ct | NPI, 1–144 |
| Avoidant attachment | Perren et al[41] | 68 | 24 m | MD 0.23 (P<0.05) | NR | |
| Expressed emotion | Vitaliano et al[44] | 77 | 15–18 m | MD 1.9 (0.77 to 3.04; P<0.001) | NR | SCB, 0–8 |
| **Outcome: psychotic symptoms (three studies)** | | | | | | |
| Mutuality | Ball et al[33] | 171 | 24 m | MD −0.1 (−0.24 to 0.04; P>0.05) (delusions) | a, g, d, e | NPI, 1–12 |
| Mutuality | Ball et al[33] | 171 | 24 m | MD −0.04 (−0.16 to 0.08; P>0.05) (hallucinations) | a, g, d, e | |
| Emotional distancing/boundary ambiguity | Caron et al[36] | 60 | 12 m | Corr 0.30 (P<0.001) (paranoia) | NR | BEHAVE-AD, 0–75 |
| Non-adaptive coping | de Vugt et al[46] | 69 | 12 m | MD NR (P>0.05) (psychosis) | g, s | NPI, 1–24 |
| **Outcome: depression (five studies)** | | | | | | |
| Mutuality | Ball et al[33] | 171 | 24 m | MD −0.43 (−0.80 to −0.06; P<0.05) | a, g, d, e | HAM-D, 0–68 |
| Quality of relationship | Burgener and Twigg[20] | 70 | 18 m | Corr 0.22 (P>0.05) | NR | Cornell, 0–38 |
| Emotional distancing/boundary ambiguity | Caron et al[36] | 60 | 12 m | Corr 0.01 (P>0.05) | NR | BEHAVE-AD, 0–75 |
| Expressed emotion | Vitaliano et al[44] | 77 | 15–18 m | MD 2.00 (−0.31 to 4.31; P>0.05) | NR | HAM-D, 0–68 |
| Cohesion quantity | Wright et al[43] | 14 | 6 m | Corr −0.302 (P>0.05) | NR | Zung, 0–270 |
| Quality | Wright et al[43] | 14 | 6 m | Corr −0.387 (P>0.05) | NR | |
| Affection quantity | Wright et al[43] | 14 | 6 m | Corr −2.41 (P>0.05) | NR | |
| Quality | Wright et al[43] | 14 | 6 m | Corr −0.038 (P>0.05) | NR | |
| Tension quantity | Wright et al[43] | 14 | 6 m | Corr 0.533 (P<0.01) | NR | |
| Quality | Wright et al[43] | 14 | 6 m | Corr −0.288 (P>0.05) | NR | |
| **Outcome: other** | | | | | | |
| Non-adaptive coping | de Vugt et al[46] | 69 | 12 m | MD NR (P=0.512) (apathy) | g, s, ct | NPI, 1–48 |
| Non-adaptive coping | de Vugt et al[46] | 69 | 12 m | MD NR (P=0.005) (hyperactivity) | g, s, ct | NPI, 1–60 |

Continued

**Table 4** Continued

| Risk factor | Study | n | Follow-up | Results (95% CI; P value) | Analyses adjusted for* | Outcome assessment tool, scale range |
|---|---|---|---|---|---|---|
| Mutuality | Morgan et al[34] and Kunik et al[32] | 171 | 24 m | MD −0.42 (−0.64 to 0.20; P<0.001) HR 0.63 (0.45 to 2.87; P=0.006) (aggression) | NR | CMAI, 0–156 |
| Emotional distancing/boundary ambiguity | Caron et al[36] | 60 | 6 m | Corr 0.15 (P>0.05) (anxiety) | NR | BEHAVE-AD, 0–75 |

*Prespecified key confounders: a, age; d, dementia severity; g, gender; s, socioeconomic status. All other confounders: ct, carer type; e, ethnicity.
HR > 1 Indicates increased risk, HR < 1 indicates decreased risk . OR > 1 indicates increased odds of outcome, OR < 1 indicates decreased odds of outcome.
BEHAVE-AD, Behavioural Pathology in Alzheimer's Disease Rating Scale; BPSD, Behavioural and Psychological Symptoms of Dementia; CMAI, Cohen Mansfield Agitation Inventory; Corr, correlation (This is a number between −1 to +1 and indicates the degree to which the exposure and outcome vary together. Positive numbers indicate that exposure and outcome increase together, negative numbers indicate that the outcome increases as the exposure decreases. Larger numbers indicate stronger correlation); Cornell, Cornell Scale for Depression in Dementia; HAM-D, Hamilton Rating Scale for Depression;MD, mean difference in outcome (Large differences between groups suggests that the exposure might affect the outcome. For continuous exposures, mean difference represents the change in the outcome for one unit increase in the exposure); NPI, Neuropsychiatric Inventory; NR, information not reported (where possible results and 95%CIs were calculated from raw data); PoV, proportion of variance (This is a number between 0–1 indicating the proportion of variance in the outcome explained by the exposure. Higher numbers indicate greater explanatory power of the exposure); SCB, Screen for Caregiver Burden (selection of items); Zung, Short Zung Interviewer Assisted Depression Rating Scale.

when reaching conclusions, thus avoiding overinterpretation of findings from studies with potentially serious problems.

Very few studies adjusted their analyses for key prespecified confounding factors (age, gender, SES and dementia severity). Two-thirds of included analyses did not adjust for any confounding factors. This is an important limitation as it means any findings could be explained by differences in these factors across the exposure groups—differences that are very likely in a non-randomised study where participants self-select into exposure groups. A small number of studies were adjusted for other factors, but these were factors such as QoL and emotional withdrawal that may well be on the causal pathway between the relationship factor studied and the outcome (so not a true confounder). Reporting of results was incomplete with many studies only reporting P values, or just that an association was 'statistically significant'/'not significant' with no effect sizes or CIs. This is another important limitation as in these cases we have no information about the probable magnitude of effect, and it is impossible to interpret the potential clinical implications of any statistical difference. Inadequate reporting of methods and results made it difficult to assess risk of bias. Very few studies evaluated the same risk factors, so there is also very little evidence on any individual factor. Duration of follow-up (typically 6–24 months), although theoretically long enough for an outcome to have occurred, may not have been sufficient to detect outcomes in the majority of the sample in a study. As dementia is typically a slowly progressing disease, this may not be sufficient for long-term outcomes such as institutionalisation.

Finally, majority of the studies identified were published before 2000. The most recent study is from 2016, with only three studies reported in the last 5 years. The reasons for this apparent diminishing of literature are unclear, but the relative absence of more recent studies may have implications for applicability of the findings of this review. New long-term studies that are well conducted and fully reported are needed to reliably answer questions regarding effect of family relationship quality on institutionalisation risks.

The current evidence does not provide a basis by which general practitioners or other health professionals could reliably identify people at risk of poor outcome on the basis of relationship factors. However, the lack of robust evidence about the role of relationship factors does not imply that personal relationships are not important factors in dementia outcomes; many professionals working with families consider these to be important.[55 56] One plausible mechanism is that factors such as emotional withdrawal of the caregiver might prompt 'challenging behaviour' in the person with dementia as an attempt to elicit an emotional connection.[41] For instance, caregiver-coping strategies such as constantly 'correcting' the person with dementia, rather than accepting their cognitive challenges, could be construed as potentially undermining the individual's personhood or self-esteem, both of which

**Table 5** Associations between relationship factors and hospitalisation, quality of life and time to death

| Risk factor | Study | n | Follow-up | Results (95% CI; P value) | Analyses adjusted for* |
|---|---|---|---|---|---|
| **Hospitalisation (one study)** | | | | | |
| Relationship strain | Godwin et al[35] | 296 | 12 m | OR 1.03 (0.92 to 1.14; 0.637) | NR |
| **Quality of life (two studies)** | | | | | |
| Quality of relationship (patient view) | Clare et al[37] | 51 | 20 m | MD 0.31 (P=0.06) | pqol, pd, cs, cqor |
| | Shroff[52] | 83 | NR | $R^2$ 0.179 (P<0.001) | NR |
| Quality of relationship (carer view) | Clare et al[37] | 51 | 20 m | MD −0.13 (P=0.89) | pqol, pd, cs, cqor |
| **Time to death (two studies)** | | | | | |
| Instrumental coping | McClendon et al[51] | 141 | 5–9 years | HR 0.99 (P=0.915) | NR |
| Acceptance coping | McClendon et al[51] | 141 | 5–9 years | HR 1.09 (P=0.644) | NR |
| Wishful thinking | McClendon et al[51] | 141 | 5–9 years | HR 1.41 (P=0.019) | NR |
| | Snyder[53] | 233 | NR | HR 0.88 (0.673 to 1.171; 0.4) | a, d, g, nc |
| Problem focused coping | Snyder[53] | 233 | NR | HR 0.80 (0.571 to 1.128; 0.2) | a, d, g, nc |
| Seeking social support | Snyder[53] | 233 | NR | HR 1.056 (0.787 to 1.416; 0.7) | a, d, g, nc |
| Blaming self | Snyder[53] | 233 | NR | HR 0.967 (0.768 to 1.218; 0.7) | a, d, g, nc |
| Avoidance coping | Snyder[53] | 233 | NR | HR1.021 (0.720 to 1.448; 0.9) | a, d, g, nc |
| Blaming others | Snyder[53] | 233 | NR | HR 0.867 (0.632 to 1.190; 0.3) | a, d, g, nc |
| Counting blessings | Snyder[53] | 233 | NR | HR 0.648 (0.454 to 0.926; 0.017) | a, d, g, nc |
| Religiosity | Snyder[53] | 233 | NR | HR 0.882 (0.682 to 1.142; 0.341) | a, d, g, nc |

*Prespecified key confounders: a, age; d, dementia severity; g, gender; s, socioeconomic status. All other confounders: cqor, carer quality of relationship; cs, carer stress; nc, non-coresidency; pd, PWD depression; Pqol, PWD quality of life.
HR > 1 indicates increased risk, HR < 1 indicates decreased risk . OR > 1 indicates increased odds of outcome, OR <1 indicates decreased odds of outcome.
MD, mean difference in outcome (Large differences between groups suggests that the exposure might affect the outcome. For continuous exposures, mean difference represents the change in the outcome for one unit increase in the exposure); NR, information not reported (where possible results and 95% CIs were calculated from raw data).

are seen as key elements in good dementia care. In this way, any 'challenging behaviour' might be construed as behavioural expressions of underlying frustration or distress. It is also possible that challenging behaviours are themselves on the causal pathway between relationship factors and institutionalisation—the latter plausibly becoming more likely as the caregiver becomes less able to cope with challenging behaviours. If this is the case, then longer follow-up may be necessary to detect associations between relationship factors and institutionalisation. In addition, the association between quality of relationships, challenging behaviour and subsequent institutionalisation is likely to be complex. A recent study[54] found that relationship quality was one of a number of psychosocial factors associated with caregiver distress at challenging behaviour independently of the frequency of that behaviour. Similarly, changes in the meaning of their relationship, and in particular the belief that their relative had lost, or would inevitably lose, their identity to dementia, is a fundamental reason why family carers experienced behaviour as challenging.[55] It may be, then,

that the quality of relationship acts as a confounding variable in the association between challenging behaviour and institutionalisation.

Lack of evidence on what relationship factors predict outcomes in people with dementia should not preclude further evaluation of psychosocial interventions targeting people with dementia and their carers. Such studies could use experimental designs to identify which interventions work and in which settings. It may also be useful to look to qualitative research, exploring the views of people with dementia and their family carers on what they consider important for continued living at home and the challenges in the relationship that they face.

## CONCLUSIONS
There is currently no strong or consistent evidence on the effects of relationship factors on institutionalisation, hospitalisation, death or QoL for people with dementia. There was a suggestion of an association between relationship factors and challenging behaviour, although

the evidence for this was weak. As the current focus of dementia care is 'person-centred', which prioritises inter-personal relationships, this lack of evidence about the role of relationships in dementia outcomes is striking. To improve our ability to support those with dementia and their families, this evidence gap needs to be addressed.

**Correction notice**  This article has been corrected since it was first published. The statement 'HBE and SI are joint first authors' has been added into the article.

**Acknowledgements**  The authors would like to thank Rebecca Beynon for support with the screening of the search records.

**Contributors**  HBE was the lead researcher, contributed to the study design, carried out screening, data extraction, data analysis and prepared the manuscript. SI carried out screening, data extraction, the risk of bias assessment and revisions of the manuscript for the review update in Summer 2017. VL and JS contributed to the screening and data extraction. JS and PW led the study design, supervised the review and contributed to drafting the manuscript. AR developed the search strategy and conducted the searches. RC and SC conceived the original study idea, and contributed to the study design. RC provided clinical advice throughout the review and made substantive critical revisions to the manuscript. All authors read and approved the final manuscript.

**Funding**  This research is supported by the National Institute for Health Research (NIHR) Collaboration for Leadership in Applied Health Research and Care West at University Hospitals Bristol NHS Foundation Trust.

**Disclaimer**  The views expressed are those of the author(s) and not necessarily those of the NHS, the NIHR or the Department of Health.

**Competing interests**  None declared.

**Patient consent**  Not required.

**Provenance and peer review**  Not commissioned; externally peer reviewed.

**Data sharing statement**  There is no additional unpublished data from this study. Any requests for access to the original data extracted from included papers should be sent to the corresponding author.

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
