## [Reviewer comments · BMJ Open]

ARTICLE DETAILS

TITLE (PROVISIONAL)	Quality of family relationships and outcomes of dementia: a systematic review
AUTHORS	Edwards, Hannah; Ijaz, Sharea; Whiting, Penny; Leach, Verity; Richards, Alison; Cullum, Sarah; Cheston, Richard; Savovic, Jelena

VERSION 1 – REVIEW

REVIEWER	George D. Papandonatos Brown University, USA.
REVIEW RETURNED	04-Jan-2017

GENERAL COMMENTS	The authors attempt to evaluate the association between the quality of relationships between persons with dementia and their caregivers on one hand and time to institutionalizations on the other. Secondary outcomes were hospitalization, death, quality of life, and challenging behavior by the person with dementia. Between-study differences precluded a quantitative synthesis of the findings, lessening the need for a formal statistical review of the paper. Still, concerns remain regarding the potential confounders under investigation and the choice of primary outcome. Although age and gender of the persons with dementia were both considered as potential confounders, less attention was given to these variable as measured in the caregiver. Further, in studies of this kind, variables defined at the dyad level might matter even more, e.g. whether a male person with dementia was being taken care by a female caregiver or vice versa; the age difference between the two; and whether they were spouses, siblings, or parents being taken care of by their children. It is possible that stratified analyses of elderly male ADs being taken care of by younger female spouses might lead to different conclusions than those of younger female ADs taken care by their male offspring. Of note, the "attachment style" construct was originally developed for parent-child dyads, and may not be as appropriate for adult romantic relationships. Given that the authors had trouble finding studies with sufficient control for AD subject characteristics, it is unlikely that they would find studies using dyads as the unit of analysis, yet such studies would more appropriate for answering the research question of interest. It also appears to this reviewer that the decision to exclude measures of caregiver burden as an exposure of interest limits the impact of the study and is seems somewhat artificial, given that "boundary ambiguity" is clearly associated with elevated caregiver burden.
---

	Finally, the authors need to clarify whether studies examining "time to institutionalization" measured "time to first institutionalization in the presence of death". Clearly, first institutionalization and death are competing risks in an AD setting, and deaths should not be treated in the same manner as administrative censoring. Did any of the studies employ a competing risks framework to deal with informative censoring due to death? Were only first event times considered for survival analyses of institutionalization outcomes? What about hospitalizations, which typically occur after institutionalization?
--	--

REVIEWER	Lian-Hua Huang, RN, EMBA, PhD, FAAN Professor, School of Nursing, College of Medicine, National Taiwan University, Taiwan
REVIEW RETURNED	12-Jan-2017

GENERAL COMMENTS	The topic is very important, as dementia is emerging as a major health issue with the growth of an ageing population worldwide. Family members are the primary source of care for loved ones with dementia. The quality of family relationships and outcome of dementia is an issue to be examined in depth, and a systematic review is appropriate method for this topic. In summary, the paper is well written. The method is clearly stated, including search strategy, selection of papers, data extraction and analysis. Risk of bias in included studies was also discussed. I would suggest accepting this manuscript for publication after minor correction the following points.  1. P8L55 "Blind independent outcome assessment was not possible in many studies with institutionalisation as the outcome as this outcome tended to be self-reported by the carer", the second "as" change to "because". Blind independent outcome assessment was not possible in many studies with institutionalisation as the outcome, because this outcome tended to be self-reported by the carer. 2. P10 Table 1:, P12 Table 2:, P14 Table 3:, P15 Table 4:, P20 Figure 1:, delete all ":". 3. P15L3 "None of the analyses for adjusted for any of the pre-specified confounding factors.", should take off the first "for". None of the analyses adjusted for any of the pre-specified confounding factors. 4. P15 Table 3 ...factors and hospitalisation; quality of life; and death, ";" should be replaced by "," ...factors and hospitalisation, quality of life, and death 5. P17L16 "our ability to answer the research question was limited due to poor reporting", please change the words "poor reporting", due to the fact research papers were done in the past, the requirement may be different at different historic period, please avoid using the term to judge the value. 6. P18L16 "this is unlikely have been sufficient for long-term outcome", rewrite as "this is not sufficient for long-term outcome".
---

REVIEWER	Georgina Charlesworth Research Department of Clinical, Educational and Health Psychology, University College London, UK
REVIEW RETURNED	20-Jan-2017

GENERAL COMMENTS	This is an interesting and important topic. The strengths of the review are: the quality of the protocol, which has been previously published; the clarity of reporting of methods and results; and the comprehensive approach to searching.  1. The major limitation of the review is the time since the searches were carried out. It is now January 2017, and the searches were undertaken in May 2015. Indeed, the literature searches (May 2015) pre-date the submission of the protocol (December 2015). An updated search would strengthen the review. 2. The identified literature is mainly pre-2000 (n=10/18) with the most recent being 2014. Given the age of the literature, it is not surprising that there are methodological and reporting limitations as older papers pre-date the introduction of methods for systematic critical appraisal of research literature. The age of the literature should be acknowledged in the discussion. 3. The size of studies and whether sample size would be adequate to identify the association of interest does not seem to have been taken into consideration. Small sample size is likely to explain the lack of statistically significant association in some studies and should be discussed. 4. Various indicators of relationship quality are mentioned in the introduction as key exposures e.g. attachment style, expressed emotion, mutuality, boundary ambiguity. However, these do not all appear in the search strategy and it is not clear whether they were identified before or after the literature was undertaken (see "when reviewing literature....p. 5, line 22). It might be that consideration of these concepts would be more appropriate in the discussion. 5. The review would benefit from inclusion of information on the tools used to measure the various dimensions of quality of relationship, including an indication of their psychometric properties. 6. Greater justification is required for why 'coping' is considered a measure of relationship quality. 'Relationship-focussed coping' has an interpersonal element, but it is not clear that instrumental coping, acceptance coping or wishful thinking are measures of relationship quality. 7. Greater justification is required for why abuse has been excluded. In the published protocol the plan was to include abuse as long as it was not in populations specifically sampled for abuse. 8. 10 potential confounders are listed in the protocol; only 4 are considered in this paper. An explanation should be provided on why the number of confounders of interest was reduced to 4. Also, clarification is should be provided as to whether the age and gender under consideration is that of the carer or person with dementia. 9. The differing perspectives of the carer and person with dementia are of note (table 4: Clare)
--

	10. The discussion would be enhanced by consideration of why the number of studies looking at the association between quality of relationship and outcomes for the person with dementia has reduced over time (if this is indeed the case following a search of the last 2 years of literature). Have other reviews considered the quality of relationship or meaning of change in relationship and presence of challenging behaviour (e.g. Feast et al., BJP, 2016). Might quality of relationship be a confounding variable in the association between challenging behaviour and institutionalisation? Quality of relationship is associated with BPSD-related distress (e.g. Feast, Int J Ger Psychiat, 2016). Typos:  • Results from 20 papers are reported not 21 – no results are reported for Kunik 2010 • Table 2: citation missing for Wright for affection & cohesion • Directing relatives behaviour – is a possessive apostrophe required as the behaviour belongs to the relatives? • Warmth and Criticism are subscales of quality of relationship for the Spruytte paper indicating double reporting • References 52 & 53. Surnames missing for Kitwood and Bender respectively. • Check psychinfo vs psycinfo throughout paper.
--	---

REVIEWER	Rónán O’Caoimh National University of Ireland, Galway - Ireland
REVIEW RETURNED	21-Feb-2017

GENERAL COMMENTS	Quality of family relationships and outcomes of dementia: a systematic review: Overall, a polished systematic review on an important topic in dementia care that sheds light on the lack of robust data in this area. Abstract and Introduction The abstract is well written and direct. A definition of what is meant by ‘quality’ should appear early on in the abstract or introduction. This is quite a subjective term and should be clarified early on in the text to avoid ambiguity. Methods While it is mentioned in the first (protocol) paper, for ease, could more detail on the type of studies included/excluded please be provided in the text. For example, that qualitative studies are excluded is important and it was only after reading the protocol that I had more certainty about the type of study included. Some specific Qs: Were data available about the quality of past relationships? Did studies adjust for caregiver burden or caregiver strain? I see that it was included as a search term in Embase and Psycinfo. Why was this not considered as an important variable/confounder to adjust for in advance like age, gender, SES and dementia severity? How was decision to include only these reached? Strain is likely to have a significant impact on the ability of carers to manage and thus on the quality of the relationship and ultimate risk of adverse outcome e.g. institutionalisation.
--

	Likewise dementia subtype would be another important confounder (e.g. frontal dementia). Please justify the reason for selecting the Newcastle-Ottawa Scale? This has its critics, particularly relating to its use in evaluating non-RCTs (see Andreas Stang). Critical evaluation of the Newcastle-Ottawa scale for the assessment of the quality of nonrandomized studies in meta-analyses. European Journal of Epidemiology, Springer Verlag, 2010, 25 (9), pp.603-605.) Results In the first paragraph clarify what is meant by participants: carers/patients/both? In Table 2, where not already clear, please add the direction of the relationship of association e.g. tension: is that low levels of tension or high levels? The authors mention and show in Table 4 how infrequently the studies adjusted for the 'a priori' selected confounders. I may be missing it but how frequently did the studies adjust for other important confounders such as those mentioned above (carer strain/dementia subtype) or others that might be relevant? The discussion and conclusion are balanced and again well written.
--	---

REVIEWER	Judith Godin Nova Scotia Health Authority and Dalhousie University Canada
REVIEW RETURNED	25-Apr-2017

GENERAL COMMENTS	Review of Quality of family relationships and outcomes of dementia: A systematic review Understanding how quality of the relationship between caregiver and person with dementia impacts outcomes such as institutionalization and quality of life is important and could lead to beneficial psychosocial interventions for caregivers and care receivers. The review that the authors present is a step in understanding this relationship. There are a number of issues that need to be address before this work is ready for publication. The biggest issue is the decision to attempt a systematic review as opposed to a scoping review. The current paper seems to be a bit of a mix between a systematic review and a scoping review. For the following reasons I think a scoping review is a better choice. First, the research question presented is much too broad for a systematic review as evidenced by the multiple factors and outcomes that are covered in the review. A systematic review should target a specific research question and all inclusion and exclusion criteria should be set clearly a priori. Two, in the protocol article the authors state that "A somewhat flexible approach will be necessary for this review as it is difficult to know in advance the nature of the studies and data that may be available". If this type of flexibility require due to a lack of knowledge regarding the current state of the literature, a scoping review is a more appropriate choice. Three, the authors state in the introduction that "it is not clear which elements of the relationships are predictive of early institutionalization or which lead to faster decline".
---

	A scoping review would help you answer this question by examining the full body of research connecting different aspects of relationship quality to institutionalization and other outcomes. A systematic review would be more appropriate to investigate the effect of a specific factor and a specific outcome (i.e., between one aspect of relationship quality and an outcome). The authors defined quality of relationship at the onset, but do not specify how they came up with this definition. The statement “Other factors emerging from the literature were included if they captured an element of relationship quality, and this was assessed on a case-by-case basis through discussion with the study team” suggests that the authors are not set on their definition of relationships quality. This is another indicator that a scoping review would be a more appropriate approach. Although there is a previously published protocol article, the submitted review should be able to stand on its own; however, the authors do not present enough information regarding their methodology for the paper to stand on its own. For instance, in the initial extraction phase how many articles were grey literature, conference proceedings etc. Did any non-English articles come up? How were these handled? What was the timing of the relationship quality – pre or post diagnosis? In the protocol article the authors discussed two sets of analyses to deal with this, but the timing is not mentioned in the submitted review. When describing results (e.g., “Six studies had inadequate follow-up”) citations for those studies should follow. The authors state in the discussion that duration of follow-up may not have been sufficient to detect outcomes but in Table 1 all but one study had a checkmark for follow-up long enough for outcome to occur. The authors need to be sure that the discussion reflects the results.
--	--

VERSION 1 – AUTHOR RESPONSE

Reviewer: 1

Reviewer Name: George D. Papandonatos

Institution and Country: Brown University, USA.

Please state any competing interests: None declared.

Please leave your comments for the authors below

Comment 1. The authors attempt to evaluate the association between the quality of relationships between persons with dementia and their caregivers on one hand and time to institutionalizations on the other. Secondary outcomes were hospitalization, death, quality of life, and challenging behavior by the person with dementia. Between-study differences precluded a quantitative synthesis of the findings, lessening the need for a formal statistical review of the paper.

Still, concerns remain regarding the potential confounders under investigation and the choice of primary outcome. Although age and gender of the persons with dementia were both considered as potential confounders, less attention was given to these variable as measured in the caregiver. Further, in studies of this kind, variables defined at the dyad level might matter even more, e.g. whether a male person with dementia was being taken care by a female caregiver or vice versa; the age difference between the two; and whether they were spouses, siblings, or parents being taken care of by their children. It is possible that stratified analyses of elderly male ADs being taken care of by younger female spouses might lead to different conclusions than those of younger female ADs taken care by their male offspring. Of note, the "attachment style" construct was originally developed for parent-child dyads, and may not be as appropriate for adult romantic relationships.

Given that the authors had trouble finding studies with sufficient control for AD subject characteristics, it is unlikely that they would find studies using dyads as the unit of analysis, yet such studies would more appropriate for answering the research question of interest.

Authors' Response: Thank you for your comment. Firstly, as we noted in the text, while children and caregivers remained the primary focus of attachment theory for many years after its original development by Ainsworth and Bowlby, in the late 1980s it was extended to adult relationships initially by Hazan and Shaver, but then by other psychologists. We appreciate that relationships between adults including between someone with dementia and their caregiver differ in many ways from relationships between children and their parents or other caregivers. The claim is not that these two kinds of relationships are identical. The claim is that the core principles of attachment theory apply across the lifespan and to both kinds of relationships. Secondly, for this review we were interested in relationship as a function of the dyad, not from the perspective of either patient or carer. We agree with you that several known and some unknown confounders may affect the association under study. We did not exclude studies on the basis of how participants were analysed (dyad or individual) or confounders adjusted, but were not able to find enough data for different subgroups and were limited in our analysis by the nature and range of studies available on the question.

Comment 2. It also appears to this reviewer that the decision to exclude measures of caregiver burden as an exposure of interest limits the impact of the study and is seems somewhat artificial, given that "boundary ambiguity" is clearly associated with elevated caregiver burden.

Authors' Response: The impact of caregiver burden has been researched extensively as a topic on its own (e.g. Eters, Goodall and Harrison, 2008; Torti et al, 2004) and merits a separate review to assess its effects. Moreover, caregiver burden has become a somewhat fraught concept within person-centred dementia research.

This is largely due to its conceptualisation of the process of care as being exclusively uni-directional i.e. one in which the behaviour of the person with dementia impacts negatively upon the carer. Consequently, scales such as the Zarit Burden Inventory have been superseded by the Sense of Competence Scale. The focus on our study on relationships that can be mutually negotiated and renegotiated meant that we favoured the concept of boundary ambiguity.

3. Finally, the authors need to clarify whether studies examining "time to institutionalization" measured "time to first institutionalization in the presence of death". Clearly, first institutionalization and death are competing risks in an AD setting, and deaths should not be treated in the same manner as administrative censoring. Did any of the studies employ a competing risks framework to deal with informative censoring due to death? Were only first event times considered for survival analyses of institutionalization outcomes? What about hospitalizations, which typically occur after institutionalization?

Authors' Response: As mentioned earlier, we were limited by what data we found. The included studies did not distinguish the effect of censoring.

Reviewer: 2

Reviewer Name: Lian-Hua Huang, RN, EMBA, PhD, FAAN

Institution and Country: Professor, School of Nursing, College of Medicine, National Taiwan University, Taiwan

Please state any competing interests: None declared

Please leave your comments for the authors below

The topic is very important, as dementia is emerging as a major health issue with the growth of an ageing population worldwide. Family members are the primary source of care for loved ones with dementia. The quality of family relationships and outcome of dementia is an issue to be examined in depth, and a systematic review is appropriate method for this topic. In summary, the paper is well written. The method is clearly stated, including search strategy, selection of papers, data extraction and analysis. Risk of bias in included studies was also discussed. I would suggest accepting this manuscript for publication after minor correction the following points.

Comment 1. P8L55 "Blind independent outcome assessment was not possible in many studies with institutionalisation as the outcome as this outcome tended to be self-reported by the carer", the second "as" change to "because". Blind independent outcome assessment was not possible in many studies with institutionalisation as the outcome, because this outcome tended to be self-reported by the carer.

Authors' Response: Thank you for your comment, we have made the change as suggested.

Comment 2. P10 Table 1:, P12 Table 2:, P14 Table 3:, P15 Table 4:, P20 Figure 1:, delete all ":".

Comment 3. P15L3 "None of the analyses for adjusted for any of the pre-specified confounding factors.", should take off the first "for". None of the analyses adjusted for any of the pre-specified confounding factors.

Comment 4. P15 Table 3 ...factors and hospitalisation; quality of life; and death, “;” should be replaced by “,” ...factors and hospitalisation, quality of life, and death

Authors' Response to comments 2, 3 and 4: Thank you for pointing this out. The changes have been made as suggested.

Comment 5. P17L16 “our ability to answer the research question was limited due to poor reporting”, please change the words “poor reporting”, due to the fact research papers were done in the past, the requirement may be different at different historic period, please avoid using the term to judge the value.

Authors' Response: We have changed the term to unclear and incomplete reporting which is a reflection of the situation we experienced.

We would like to point out that The CONSORT statement has been around for over 20 years now, and STROBE statement for 10 years. Reporting practice should have improved in the past ten years at least. But even before, reporting mean values without variance measures and effects without confidence intervals was not considered good practice in scientific reports - a situation seen in many of the included publications.

6. P18L16 “this is unlikely have been sufficient for long-term outcome”, rewrite as “this is not sufficient for long-term outcome”.

Authors' Response: Thank you, we have made the suggested change. Page 22 line 5:

“Duration of follow-up (typically 6-24 months), although theoretically long enough for an outcome to have occurred, may not have been sufficient to detect outcomes in the majority of the sample in a study. As dementia is typically a slowly progressing disease this may not be sufficient for long-term outcomes such as institutionalisation.”

Reviewer: 3

Reviewer Name: Georgina Charlesworth

Institution and Country: Research Department of Clinical, Educational and Health Psychology, University College London, UK

Please state any competing interests: None declared

Please leave your comments for the authors below

This is an interesting and important topic. The strengths of the review are: the quality of the protocol, which has been previously published; the clarity of reporting of methods and results; and the comprehensive approach to searching.

Comment 1. The major limitation of the review is the time since the searches were carried out. It is now January 2017, and the searches were undertaken in May 2015. Indeed, the literature searches (May 2015) pre-date the submission of the protocol (December 2015). An updated search would strengthen the review.

Authors' Response: Thank you for your comment. This has been addressed now, update search carried out 17 May 2017.

Comment 2. The identified literature is mainly pre-2000 (n=10/18) with the most recent being 2014. Given the age of the literature, it is not surprising that there are methodological and reporting limitations as older papers pre-date the introduction of methods for systematic critical appraisal of research literature. The age of the literature should be acknowledged in the discussion.

Authors' Response: We have now added P22, para 2:

"Finally, majority of the studies identified were published before 2000. The most recent study is from 2016, with only three studies reported in the last five years. The reasons for this apparent diminishing of literature are unclear, but the relative absence of more recent studies may have implications for applicability of the findings of this review. New long-term studies that are well conducted and fully reported are needed to reliably answer questions regarding effect of family relationship quality on institutionalisation risks."

Comment 3. The size of studies and whether sample size would be adequate to identify the association of interest does not seem to have been taken into consideration. Small sample size is likely to explain the lack of statistically significant association in some studies and should be discussed.

Authors' Response: Appropriateness of sample and its size is an item addressed as part of the risk of bias assessment-. We have added following sentence at the end of para 2 Page 16:

"This could also be because the sample was too small to detect a difference. No study reported justification for their sampling or the sample sizes."

Comment 4. Various indicators of relationship quality are mentioned in the introduction as key exposures e.g. attachment style, expressed emotion, mutuality, boundary ambiguity. However, these do not all appear in the search strategy and it is not clear whether they were identified before or after the literature was undertaken (see "when reviewing literature....p. 5, line 22). It might be that consideration of these concepts would be more appropriate in the discussion.

Authors' Response: We included 'expressed adj2 emotion' and 'attachment adj 2 styles' in our search strategy. However we did not include 'boundary ambiguity' and 'mutuality' in the same way. This could have missed studies that only reported this aspect, although we screened for these items in abstracts and full texts. To address your comment, we carried out a sensitivity check using these as search terms. We found that the index headings of all studies reporting the aspect of boundary ambiguity alone also used 'family relations' and/or 'caregivers' in their abstracts or keywords – we used both of these terms in our search. We also identified all of these studies using the text words in our search for a) the signs and symptoms of caregiver/ dementia patient relationship (stress, distress, depression etc) and b) relationship quality (loving, dysfunctional). Thus studies picked up from our existing search should already include all those reporting the above listed items.

Comment 5. The review would benefit from inclusion of information on the tools used to measure the various dimensions of quality of relationship, including an indication of their psychometric properties.

Authors' Response: Thank you for your comment. Unfortunately, we had not extracted this information during our data extraction and with the need to carry out the full update of the review we had no capacity to go back and re-extract this from all studies. Although this information may have been useful to some readers we didn't feel it was crucial for the review, given the additional workload and limited team capacity. But most importantly, it would not have changed the findings or the conclusions of the review.

Comment 6. Greater justification is required for why 'coping' is considered a measure of relationship quality. 'Relationship-focussed coping' has an interpersonal element, but it is not clear that instrumental coping, acceptance coping or wishful thinking are measures of relationship quality.

Author Response: We agree that greater clarity about the selection of coping as an exposure is required, and have now inserted a passage into the text on p5 to clarify our reasoning for this: "Coping is a wide-ranging construct that includes elements which are clearly measures of relationship quality (e.g. 'relationship-focussed coping') and those which are more individual in nature (e.g. acceptance coping). However, as even individual coping styles are typically initiated in response to aspects of relationships, we felt that this was an appropriate exposure to capture." In Table 2, we have also differentiated between "Factors relating to the interaction between the person with dementia and their caregiver" and "Factors mainly relating to the caregiver". We have included approach coping in the latter category.

Comment 7. Greater justification is required for why abuse has been excluded. In the published protocol the plan was to include abuse as long as it was not in populations specifically sampled for abuse.

Authors' Response: We explained in our protocol that we would include studies if they included abused along with non-abused participants, however no studies of this nature of sampling were found. On page 3 paragraph 2 of our protocol publication we stated: "Studies in which participants in abusive relationships are included alongside participants in non-abusive relationships, which are also exploring our specified eligible exposures (and outcomes) of interest, will be included (eg, studies comparing risk of institutionalisation in participants in abusive relationships vs those in non-abusive relationships)."

Comment 8. 10 potential confounders are listed in the protocol; only 4 are considered in this paper. An explanation should be provided on why the number of confounders of interest was reduced to 4. Also, clarification is should be provided as to whether the age and gender under consideration is that of the carer or person with dementia.

Authors' Response: In our protocol publication we put together a more comprehensive list of confounders that we thought were relevant for studies in this field. It does not necessarily mean that each study should have adjusted for all of these confounders. However, for the purposes of assessing risk of bias we needed an essential, core list of the most important confounders that we expected each study to have accounted for. So, for the purposes of our risk of bias assessment, the study could only be judged low risk for confounding if they adjusted for all 4 of these core confounders. This does not mean that the other six are not important but not all of them would necessarily need to be adjusted for in each study. These four factors were chosen during the piloting, by team consensus, guided by of subject experts (Cheston and Cullum). We have added the following text in the Methods to clarify this:

"To be assessed as adequate for comparability of cohorts (risk of confounding) we chose a minimum set of 4 essential items based on expert consensus for the purposes of risk of bias assessment. A study had to control for the following four pre-specified factors to be at low risk of confounding: age, gender, socio-economic status (SES) and dementia severity."

Comment 9. The differing perspectives of the carer and person with dementia are of note (table 4: Clare)

Authors' response: This is interesting, but we did not want to overinterpret this finding as it is a single result from a small study of 51 participants and no confidence intervals were preported. We also tried not to repeat in the text what was already in the tables, and in the Discussion we tried to reflect on overall evidence rather than single out individual studies.

Comment 10. The discussion would be enhanced by consideration of why the number of studies looking at the association between quality of relationship and outcomes for the person with dementia has reduced over time (if this is indeed the case following a search of the last 2 years of literature). Have other reviews considered the quality of relationship or meaning of change in relationship and presence of challenging behaviour (e.g. Feast et al., BJP, 2016). Might quality of relationship be a confounding variable in the association between challenging behaviour and institutionalisation? Quality of relationship is associated with BPSD-related distress (e.g. Feast, Int J Ger Psychiat, 2016).

Authors' response: We agree with the reviewer that it is interesting to note that the number of studies reporting in this area has reduced considerably in recent years. This also seems to be true for other reviews which have considered similar themes - including those of Feast et al, which are referred to here. However, we are not aware of any evidence as to why this may be, and while we could comment on possible causes (e.g. a shifting of the research frame to other priorities) this would be entirely speculative. Moreover, we feel that it is more relevant simply to highlight the fact that the relative absence of recent evidence limits the applicability of the review.

We also agree that it is possible that quality of relationships might mediate the relationship between challenging behaviour and institutionalisation, and we have now included this possibility in our discussion section on page 23 as follows:

“In addition, the association between quality of relationships, challenging behaviour and subsequent institutionalisation is likely to be complex. A recent study found that relationship quality was one of a number of psychosocial factors associated with caregiver distress at challenging behaviour independently of the frequency of that behaviour.

Similarly changes in the meaning of their relationship, and in particular the belief that their relative had lost, or would inevitably lose, their identity to dementia, is a fundamental reason why family carers experienced behaviour as challenging. It may be, then, that the quality of relationship acts as a confounding variable in the association between challenging behaviour and institutionalisation.”

Typos:

- 1) Results from 20 papers are reported not 21 – no results are reported for Kunik 2010
- 2) Table 2: citation missing for Wright for affection & cohesion
- 3) Directing relatives behaviour – is a possessive apostrophe required as the behaviour belongs to the relatives?
- 4) Warmth and Criticism are subscales of quality of relationship for the Spruytte paper indicating double reporting
- 5) References 52 & 53. Surnames missing for Kitwood and Bender respectively.
- 6) Check psychinfo vs psycinfo throughout paper.

Authors' Response to points 1 to 6:

- 1) Kunik, Ball, Morgan and Godwin all reported the same study. Kunik paper was initially not reported as the same measure (mutuality) was reported under Morgan 2013. However, having looked at both sets of results again we thought it would be useful to also report the Hazard ratio result reported in Kunik 2010, as it is easier to interpret than mean difference.
- 2) Thank you, the citation for Wright et al has been added now.
- 3) Thank you for pointing this out - the apostrophe has been added.

4) We have extracted results for any or all components of the relationship quality as well as overall quality of relationship. Since these results were not included in a meta-analysis, the reporting of components does not lead to double-counting. We think it is useful for the reader to see the findings for the components as well for overall quality of relationship. We have however added a footnote in the table legend to make it clear that these are the subscales of the relationship quality scale.

5) This has now been amended

6) We could not find any psychinfo in the text. There is only PsycInfo which is a database and does not need changing.

Reviewer: 4

Reviewer Name: Rónán O'Caoimh

Institution and Country: National University of Ireland, Galway - Ireland

Please state any competing interests: None declared

Please leave your comments for the authors below

Quality of family relationships and outcomes of dementia: a systematic review:

Comment 1) Overall, a polished systematic review on an important topic in dementia care that sheds light on the lack of robust data in this area.

Authors' Response: Thank you.

Comment 2) Abstract and Introduction

The abstract is well written and direct. A definition of what is meant by 'quality' should appear early on in the abstract or introduction. This is quite a subjective term and should be clarified early on in the text to avoid ambiguity.

Authors' Response: Thank you. We have added a definition of relationship quality at the end of the introductory section (page 4, paragraph 2):

"We adopted a broad definition of "relationship quality" as how happy or satisfied an individual is in his or her relationship."

Comment 3) Methods

While it is mentioned in the first (protocol) paper, for ease, could more detail on the type of studies included/excluded please be provided in the text. For example, that qualitative studies are excluded is important and it was only after reading the protocol that I had more certainty about the type of study included.

Authors' Response: Thank you for your suggestion. We have added a new sentence to page 4, end of paragraph 3:

"Qualitative, case control (unless nested in a prospective cohort), and cross-sectional studies were excluded."

Some specific Qs:

Comment 4) Were data available about the quality of past relationships?

Authors' Response: No, these data were not available.

Comment 5) Did studies adjust for caregiver burden or caregiver strain? I see that it was included as a search term in Embase and Psycinfo. Why was this not considered as an important variable/confounder to adjust for in advance like age, gender, SES and dementia severity? How was decision to include only these reached? Strain is likely to have a significant impact on the ability of carers to manage and thus on the quality of the relationship and ultimate risk of adverse outcome e.g. institutionalisation.

Likewise dementia subtype would be another important confounder (e.g. frontal dementia).

Authors' Response: We agree that caregiver burden or strain as well as type of dementia may well have acted as a confounder as could several others. However, we chose a minimum set of 4 essential items for the purposes of risk of bias assessment. The selection of these 4 factors was made during our pilot and based the opinion of clinical experts on our team. We have now added to the tables a list of all confounders that were adjusted for.

Comment 6) Please justify the reason for selecting the Newcastle-Ottawa Scale? This has its critics, particularly relating to its use in evaluating non-RCTs (see Andreas Stang. Critical evaluation of the Newcastle-Ottawa scale for the assessment of the quality of nonrandomized studies in meta-analyses. European Journal of Epidemiology, Springer Verlag, 2010, 25 (9), pp.603-605.)

Authors' Response: We are aware of and agree with the reviewer on the limitations of Newcastle-Ottawa Scale (NOS). However, similar criticism could be expressed for any other risk of bias or quality assessment tool for exposure studies that is currently available. There is currently no universally accepted, ideal tool to assess risk of bias in non-randomised studies. NOS is possibly one of the more widely used tools, and when we piloted it seemed to work for our review, so that's why we chose it. To strengthen the validity of our risk of bias assessment, we did not pool scores of quality and have carried out domain based assessment using NOS items, as adding up scores is generally considered bad practice in risk of bias assessment.

The much more comprehensive and rigorously developed ROBINS-I (Risk of Bias In Nonrandomised Studies of Interventions) tool is not suitable for exposure studies. Its sister tool, ROBINS- E (E = exposures) (Higgins, University of Bristol, personal communication) is under development and once this tool becomes available we will be able to use that for future reviews.

Results

Comment 7) In the first paragraph clarify what is meant by participants: carers/patients/both?

Authors' Response: Thank you that has been done now where 'participants with dementia' has replaced 'participants'.

Comment 8) In Table 2, where not already clear, please add the direction of the relationship of association e.g. tension: is that low levels of tension or high levels?

Authors' Response: We have altered Table 2 to clarify these points.

Comment 9) The authors mention and show in Table 4 how infrequently the studies adjusted for the 'a priori' selected confounders. I may be missing it but how frequently did the studies adjust for other important confounders such as those mentioned above (carer strain/ dementia subtype) or others that might be relevant?

Authors' Response: In the Results tables, we had previously reported for each included result which of the four key confounders they adjusted for, as well as whether they adjusted for any other confounders. The latter was denoted with a letter 'o' (=other confounders adjusted for) in the tables. We have now replaced 'other confounders' category with a full list of confounders controlled for (each denoted with a letter(s) and explained in the legend).

Comment 10) The discussion and conclusion are balanced and again well written.

Authors' Response: Thank you.

Reviewer: 5

Reviewer Name: Judith Godin

Institution and Country: Nova Scotia Health Authority and Dalhousie University, Canada

Please state any competing interests: None declared.

Please leave your comments for the authors below

Review of Quality of family relationships and outcomes of dementia: A systematic review
Understanding how quality of the relationship between caregiver and person with dementia impacts outcomes such as institutionalization and quality of life is important and could lead to beneficial psychosocial interventions for caregivers and care receivers. The review that the authors present is a step in understanding this relationship.

Comment 1) There are a number of issues that need to be address before this work is ready for publication. The biggest issue is the decision to attempt a systematic review as opposed to a scoping review. The current paper seems to be a bit of a mix between a systematic review and a scoping review. For the following reasons I think a scoping review is a better choice.

Authors' Response. We do not agree that this systematic review is anything other than a full a-priori designed systematic review. We present our arguments next to each of the reviewer's below.

Comment 2) First, the research question presented is much too broad for a systematic review as evidenced by the multiple factors and outcomes that are covered in the review. A systematic review should target a specific research question and all inclusion and exclusion criteria should be set clearly a priori.

Authors' Response: Having a specific question defined (in the case of risk factor reviews by specifying patients, exposures, and if necessary, outcomes) and study designs included is a requirement of a systematic review as well as a scoping review. The key difference is in their aims, which affects the study types that may be included and how the studies will be handled. A scoping review compared to a systematic review 1) addresses a broad topic or area of research aiming to describe the range of research activity, rather than an estimate of an effect (of exposure or intervention) 2) does not require a quality assessment and 3) does not require a synthesis of study findings.

The differences are outlined by Arksey and O'Malley (Arksey and O'Malley 2005)– an accepted standard in scoping review methods- as:

“So what might we consider to be the main differences between a systematic review and a scoping study? First, a systematic review might typically focus on a well-defined question where appropriate study designs can be identified in advance whilst a scoping study tends to address broader topics where many different study designs might be applicable.

Second, the systematic review aims to provide answers to questions from a relatively narrow range of quality assessed studies, whilst a scoping study is less likely to seek to address very specific research questions nor, consequently, to assess the quality of included studies.... The scoping study does not address the issue of 'synthesis', that is the relative weight of evidence in favour of the effectiveness of any particular intervention. Consequently, scoping studies provide a narrative or descriptive account of available research."

A Cochrane systematic review can be broad in terms of either participants, interventions, comparisons or outcomes (Cochrane Handbook, part 2, Section 5.6; <http://handbook-5-1.cochrane.org/>). Having multiple outcomes and/or interventions does not mean it as a scoping review. (Wiles, Williams et al. 2013, Murphy, Froggatt et al. 2016, Shrestha, Kukkonen-Harjula et al. 2016) We acknowledge that this is not an effectiveness of interventions review. However, exposure factors/ risk factors, prognostic factors and diagnostic methods have all been addressed in systematic reviews for the past few decades following the systematic review methods successfully to advance practice and research (Ablitt, Jones et al. 2009, Kelley, Kraft-Todd et al. 2014, Moran, Van Cauwenberg et al. 2014, Rose, Adams et al. 2016).

Comment 3) Two, in the protocol article the authors state that "A somewhat flexible approach will be necessary for this review as it is difficult to know in advance the nature of the studies and data that may be available". If this type of flexibility require due to a lack of knowledge regarding the current state of the literature, a scoping review is a more appropriate choice.

Authors' Response: The flexibility referred to is something advised in the Cochrane handbook (Part 2, chapter 5, section 7). It allows for flexibility even within the question itself as long as it has merit, and bias is minimised in such decisions by being explicit. Indeed many Cochrane review protocols anticipate a lack of evidence or limitations of knowledge (hence the need for a review) and state clearly that they would consider changing inclusion thresholds if compelling reasons exist. The explicit reporting of such decisions in completed reviews is what guards against bias or misinterpretation. Contrary to systematic reviews, scoping reviews (Arksey and O'Malley 2005) are not per se aiming to come to conclusions about (exposure or intervention) factors of interest (no synthesis narrative or otherwise is usually done). They aim to describe the current extent of knowledge in a field to identify gaps in literature for future research. Whereas systematic reviews are mainly intended to update and inform practice. Thus it is the aim what matters, not what is found. For this reason, scoping reviews also are not expected to assess risk of bias in the found literature.

Comment 4) Three, the authors state in the introduction that "it is not clear which elements of the relationships are predictive of early institutionalization or which lead to faster decline". A scoping review would help you answer this question by examining the full body of research connecting different aspects of relationship quality to institutionalization and other outcomes. A systematic review would be more appropriate to investigate the effect of a specific factor and a specific outcome (i.e., between one aspect of relationship quality and an outcome).

Authors' Response: We disagree that a scoping review could answer this question. Good quality data from robust studies with long-enough follow up, assessing a-priori hypotheses of relationship quality would help to answer this question. It is unfortunate that not enough evidence of high quality was found to provide an answer in the current review. A scoping review, without synthesis of the evidence located, would only have described what factors have been studied. It would have included all types of literature to find what hasn't been studied but we would not be any closer to the answer than we are. We also disagree that one factor studied in the review or one outcome would have produced any further clarity, because it is clear from our results that the high quality primary evidence on any one of the factors is lacking.

Comment 5) The authors defined quality of relationship at the onset, but do not specify how they came up with this definition. The statement “Other factors emerging from the literature were included if they captured an element of relationship quality, and this was assessed on a case-by-case basis through discussion with the study team” suggests that the authors are not set on their definition of relationships quality. This is another indicator that a scoping review would be a more appropriate approach.

Authors’ response: We have added a sentence in the Background section explaining that we adopted a broad definition of “relationship quality” (page 4, paragraph2):

“We adopted a broad definition of “relationship quality” as how happy or satisfied an individual is in his or her relationship.”

This team decision on definitions is a standard part of early systematic review processes, whereas in scoping reviews this (expert consensus) is advised to be a final consultative (albeit optional) exercise to enhance results and make them more relevant to a context, policy or future research (Arksey and O'Malley 2005)

Comment 6) Although there is a previously published protocol article, the submitted review should be able to stand on its own; however, the authors do not present enough information regarding their methodology for the paper to stand on its own. For instance, in the initial extraction phase how many articles were grey literature, conference proceedings etc. Did any non-English articles come up? How were these handled? What was the timing of the relationship quality – pre or post diagnosis? In the protocol article the authors discussed two sets of analyses to deal with this, but the timing is not mentioned in the submitted review.

When describing results (e.g., “Six studies had inadequate follow-up”) citations for those studies should follow.

Authors’ Response: Thank you for pointing this out. We agree that it is easier for a reader to have all information within the review rather than just the protocol. We added in the Methods that we did not apply any language restrictions. Working at a very international university we usually deal with foreign language papers by getting help from colleagues or students who a native speaker of the language of the paper. We even have a voluntary list of colleagues who offer to screen papers in their native language. However, almost all abstracts these days tend to be in English (even when the full paper is in a non-English language) and on this occasion all papers assessed in full were in English. We did not feel it was necessary to report this level of detail in the paper.

We added the missing six references. Thank you for spotting this.

Comment 7) The authors state in the discussion that duration of follow-up may not have been sufficient to detect outcomes but in Table 1 all but one study had a checkmark for follow-up long enough for outcome to occur. The authors need to be sure that the discussion reflects the results.

Authors’ Response: The term follow-up long enough refers to the minimum threshold in the Newcastle Ottawa Scale (NOS) for assessing risk of bias. This item of NOS checks that the study allowed for at least a minimum plausible length of follow-up from exposure to when the outcome might occur. However, in many studies the majority of patients might experience outcome much later than the earliest likely time point, which would lead to small number of events and thus low power to detect an association. We have now rephrased our Discussion to reflect this (page 22, end of 1st paragraph):

“Duration of follow-up (typically 6-24 months), although theoretically enough for an outcome to have occurred, may not have been sufficient to detect outcomes in the majority of the sample in a study. As dementia is typically a slowly progressing disease this may not be sufficient for long-term outcomes such as institutionalisation.”

References

- Ablitt, A., G. V. Jones and J. Muers (2009). "Living with dementia: A systematic review of the influence of relationship factors." *Aging & Mental Health* 13(4): 497-511.
- Arksey, H. and L. O'Malley (2005). "Scoping studies: towards a methodological framework." *International Journal of Social Research Methodology* 8(1): 19-32.
- Kelley, J. M., G. Kraft-Todd, L. Schapira, J. Kossowsky and H. Riess (2014). "The Influence of the Patient-Clinician Relationship on Healthcare Outcomes: A Systematic Review and Meta-Analysis of Randomized Controlled Trials." *PLOS ONE* 9(4): e94207.
- Moran, M., J. Van Cauwenberg, R. Hercky-Linnewiel, E. Cerin, B. Deforche and P. Plaut (2014). "Understanding the relationships between the physical environment and physical activity in older adults: a systematic review of qualitative studies." *International Journal of Behavioral Nutrition and Physical Activity* 11(1): 79.
- Murphy, E., K. Froggatt, S. Connolly, E. O'Shea, E. L. Sampson, D. Casey and D. Devane (2016). "Palliative care interventions in advanced dementia." *Cochrane Database of Systematic Reviews*(12).
- Rose, T. C., N. Adams, D. C. Taylor-Robinson, B. Barr, J. Hawker, S. O'Brien, M. Violato and M. Whitehead (2016). "Relationship between socioeconomic status and gastrointestinal infections in developed countries: a systematic review protocol." *Systematic Reviews* 5(1): 13.
- Shrestha, N., K. T. Kukkonen-Harjula, J. H. Verbeek, S. Ijaz, V. Hermans and S. Bhaumik (2016). "Workplace interventions for reducing sitting at work." *Cochrane Database Syst Rev* 3: CD010912.
- Wiles, N., C. J. Williams, D. Kessler and G. Lewis (2013). "Psychological therapies for treatment-resistant depression in adults." *Cochrane Database of Systematic Reviews*(6).

VERSION 2 – REVIEW

REVIEWER	George Papandonatos Brown University, USA
REVIEW RETURNED	21-Sep-2017

GENERAL COMMENTS	Although the authors are correct that the lack of adjustment for a common set of confounders make the various measures of association hard to combine across studies, they could have done more to harmonize these measures themselves and convert them to effect sizes. For example, the proportion of variance explained in a univariate normal regression model is nothing but the square of the correlation coefficient between the single predictor and the outcome. If the direction of the association is reported in the paper, the POV finding can be converted to a (signed) correlation coefficient. Similarly, if the correlation coefficient is not reported, but the authors include the value or p-value of a T or Chi-square statistic as well as its degrees of freedom, then one can again recover the correlation from the data. One website calculator that automates such conversions is given at http://www.polyu.edu.hk/mm/effectsizefaqs/calculator/calculator.html, but many other such relationships can be found online. The authors are encouraged to do this, before concluding that the associations measures of interest are too diverse to allow meta-analytic synthesis.
---

REVIEWER	Lian-Hua Huang School of Nursing, College of Medicine, National Taiwan University Taiwan
REVIEW RETURNED	11-Sep-2017

GENERAL COMMENTS	 1. P19L7 one extra "." should be deleted 2. P14 Table 2: → Table 2. 3. P16 Table 3: → Table 3. 4. P18 Table 4: → Table 4. 5. P19 Table 5: → Table 5. 6. P24 Figure 1: → Figure 1.
--

REVIEWER	Rónán O'Caomh National University of Ireland, Galway, Ireland.
REVIEW RETURNED	12-Sep-2017

GENERAL COMMENTS	Thank you for making the required edits. The paper is now much improved.
--

REVIEWER	Judith Godin Nova Scotia Health Authority and Dalhousie University
REVIEW RETURNED	26-Sep-2017

GENERAL COMMENTS	Thank you to the authors for their thoughtful and thorough responses to my comments. The inclusion of a definition of relationship quality and the justification for a systematic review provided by the authors in their response have satisfied my concerns over the choice of review type. In the original submission, the lack of specified definition of relationship quality in conjunction with the fluidity of inclusion criteria (in terms of measured exposure), left me with the sense that the definition of relationship quality was somewhat "up in the air", and I thought a scoping review would be better to enable the authors to survey the concepts that make up relationship quality in this context.
---

VERSION 2 – AUTHOR RESPONSE

Reviewer: 1

Reviewer Name: George Papandonatos

Institution and Country: Brown University, USA

Please state any competing interests: None declared

Please leave your comments for the authors below:

Comment : Although the authors are correct that the lack of adjustment for a common set of confounders make the various measures of association hard to combine across studies, they could have done more to harmonize these measures themselves and convert them to effect sizes. For example, the proportion of variance explained in a univariate normal regression model is nothing but the square of the correlation coefficient between the single predictor and the outcome. If the direction of the association is reported in the paper, the POV finding can be converted to a (signed) correlation coefficient. Similarly, if the correlation coefficient is not reported, but the authors include the value or p-value of a T or Chi-square statistic as well as its degrees of freedom, then one can again recover the correlation from the data. One website calculator that automates such conversions is given at <http://www.polyu.edu.hk/mm/effectsizefaqs/calculator/calculator.html>, but many other such relationships can be found online. The authors are encouraged to do this, before concluding that the associations measures of interest are too diverse to allow meta-analytic synthesis.

Author Response: Thank you for your suggestion. We agree that standardising the effect measures for a meta-analysis is possible. However, our decision not to carry out a meta-analysis was not based on differences in statistics reported in the studies. We think that a meta-analysis would not be appropriate because of the clinical heterogeneity between studies and the varied sets of (or lack of) confounder adjustment across the studies. There were at most two studies that measured the same combination of exposure and outcome and often these were measured in different ways and differed in how the authors dealt with confounding. For example, for the outcome of institutionalisation, the 'relationship quality' exposure was measured in one study (Pruchno et al) with an author-defined question on quality of current relationship with the spouse, and in the other (Spruytte et al.) by quality of the premorbid relationship. While the latter study adjusted for twelve confounders the earlier did not report any. Their measurement of outcome (institutionalisation) is also at different lengths of follow up. So a meta- analysis, although doable is not advisable.

Reviewer: 2

Reviewer Name: Lian-Hua Huang

Institution and Country: School of Nursing, College of Medicine, National Taiwan University, Taiwan

Please state any competing interests: None declared

Please leave your comments for the authors below:

1. P19L7 one extra "." should be deleted
2. P14 Table 2: → Table 2.
3. P16 Table 3: → Table 3.
4. P18 Table 4: → Table 4.
5. P19 Table 5: → Table 5.
6. P24 Figure 1: → Figure 1.

Author Response: Thank you, we have changed these as suggested.

Reviewer: 4

Reviewer Name: Rónán O'Caomh

Institution and Country: National University of Ireland, Galway, Ireland.

Please state any competing interests: None declared.

Please leave your comments for the authors below

Thank you for making the required edits. The paper is now much improved.

Author Response: Thank you, for your comments.

Reviewer: 5

Reviewer Name: Judith Godin

Institution and Country: Nova Scotia Health Authority and Dalhousie University

Please state any competing interests: 'None declared'

Please leave your comments for the authors below

Comment: Thank you to the authors for their thoughtful and thorough responses to my comments.

The inclusion of a definition of relationship quality and the justification for a systematic review provided by the authors in their response have satisfied my concerns over the choice of review type.

In the original submission, the lack of specified definition of relationship quality in conjunction with the fluidity of inclusion criteria (in terms of measured exposure), left me with the sense that the definition of relationship quality was somewhat "up in the air", and I thought a scoping review would be better to enable the authors to survey the concepts that make up relationship quality in this context.

Author Response: Thank you, for your comments. We are pleased that you are now satisfied that our decision to do a systematic review was reasonable.

VERSION 3 – REVIEW

REVIEWER	George Papandonatos Brown University, USA
REVIEW RETURNED	18-Oct-2017
GENERAL COMMENTS	I now understand why the authors did not bother to convert the reported effect sizes to a common scale. I agree that quantitative synthesis would not have made much sense in this case.